# Molecular Dynamics and Chain Length of Edible Oil Using Low-Field Nuclear Magnetic Resonance

**DOI:** 10.3390/molecules28010197

**Published:** 2022-12-26

**Authors:** Zijian Jia, Can Liang

**Affiliations:** 1School of Health Science and Engineering, University of Shanghai for Science and Technology, 516 Jungong Road, Shanghai 200093, China; 2School of Civil Engineering & Architecture, Changzhou Institute of Technology, 666 Liaohe Road, Changzhou 213032, China

**Keywords:** low-field NMR, two-dimensional nuclear magnetic resonance, edible oil, chain length, rotation correlation time

## Abstract

Nuclear magnetic resonance (NMR) techniques are widely used to identify pure substances and probe protein dynamics. Edible oil is a complex mixture composed of hydrocarbons, which have a wide range of molecular size distribution. In this research, low-field NMR (LF-NMR) relaxation characteristic data from various sample oils were analyzed. We also suggest a new method for predicting the size of edible oil molecules using LF-NMR relaxation time. According to the relative molecular mass, the carbon chain length and the transverse relaxation time of different sample oils, combined with oil viscosity and other factors, the relationship between carbon chain length and transverse relaxation time rate was analyzed. Various oils and fats in the mixed fluid were displayed, reflecting the composition information of different oils. We further studied the correlation between the rotation correlation time and the molecular information of oil molecules. The molecular composition of the resulting fluid determines its properties, such as viscosity and phase behavior. The results show that low-field NMR can obtain information on the composition, macromolecular aggregation and molecular dynamics of complex fluids. The measurements of grease in the free-fluid state show that the relaxation time can reflect the intrinsic properties of the fluid. It is shown that the composition characteristics and states of complex fluids can be measured using low-field nuclear magnetic resonance.

## 1. Introduction

Fat plays an irreplaceable role in human diet; it gives food a unique flavor and color. The quality of edible oil is also a big issue related to the dietary health of consumers all over the world and may also endanger the health of consumers. Therefore, it is of profound significance to evaluate the quality change of edible oil during use and storage. Using traditional physical and chemical methods to evaluate the quality of edible oil is time-consuming, laborious and has large errors. Compared with traditional detection methods, nuclear magnetic resonance (NMR) is a non-destructive detection method that can maintain the integrity of samples. NMR is a powerful tool for qualitative and quantitative analysis of organic and inorganic substances. Low-field nuclear magnetic resonance (LF-NMR) technology is a new technical means to observe and analyze the physical parameters of samples that has been gradually developed in recent years. At the same time, it symbolizes the research direction of medical detection technology, and complex high-end molecular and chemical materials are being developed for applications in a wider range of industrial and agricultural fields. NMR technology can be used to observe and analyze the characteristics of a material without damaging the sample, with the advantages of being fast, accurate and non-invasive, with no pollution, no radiation and so on. High-field NMR equipment has high sensitivity, high resolution and a high signal-to-noise ratio, but it has high requirements for sample uniformity. Liquids need to be deionized and solids need to be powdered. In addition, the instrument is expensive, and the cost of subsequent maintenance equipment is extremely high. LF-NMR devices, however, use permanent magnets, making them small and cheap. They are thus ideal for online process inspection, industrial quality control and quality inspection. Compared with high-field NMR, LF-NMR is both inexpensive and rapid. Low-field NMR instruments can be made portable. Testing can be done on site rather than in a fixed laboratory [1].

Mixed porous media composed of water and oil are present in most food materials, and the relaxation rate of samples is affected by the pore size. Therefore, the measurement of pore distribution and the distinction between oil and water can be determined and analyzed using LF-NMR. This can also play an important role in the quality control of food safety, such as edible oil [2,3,4,5], yellow croaker [6], prawn [7,8], etc. The transverse relaxation decay curve of LF-NMR can not only be used to monitor the quality of frying oil based on the prediction of physicochemical indices, such as viscosity, acid value and carbonyl value [9], but also to elucidate the correlation between the biophysical state of pure water and the structural properties in dried fermented meat products [10]. What is more, the analysis of transverse relaxation decay curves also plays an important role in food quality control. It has been widely used in the analytical detection of food products such as edible oil incorporation, artifact detection [11], soybean variety discrimination [12] and meat quality analysis [13].

1H NMR spectroscopy has been widely used in the analysis of oil to determine vegetable oil bio-sources [14] and vegetable oil fatty acid profiles [15]. Techniques based on 1H NMR spectroscopy have been intensively developed to characterize bio-oil composition, monitor the bio-oil production process and evaluate the bio-oil concentration in bio-oils and their mixtures [16]. The determination of carbon chain length is critical for liquid fuels because it is one of the key parameters determining their quality and performance [17,18]. Additionally, it strongly influences other quality parameters, such as the viscosity of liquid fuels, thermal values and hexadecane values. The correlation of experimentally obtained *T*_2_ relaxation times with viscosity and other physical parameters of different kinds of vegetable oil and other oils is empirically described in most studies. NMR relaxation is used to explain the *T*_2_ dependence on the viscosity of alkanes, considering the shape of spherical molecules [19].

In this work, we applied low-field NMR to study the molecular structures of fatty acids and glycerides. The correlation between relaxation rates and the carbon chain lengths of fatty acids and organic fluids suggests that low-field NMR has the potential to serve as a method for rapidly measuring the properties of oily moieties.

## 2. NMR Theory

Molecular motion determines the relaxation time and available information about molecular components. The spin dynamics of a fluid are characterized by the longitudinal relaxation time *T*_1_, transverse relaxation time *T*_2_ of the spin system and the diffusion coefficient D of the whole molecule. Among relatively complex fluid mixtures, small molecules diffuse more rapidly than large ones due to their molecular volume. Thus, the diffusion coefficient of a particular hydrocarbon molecule is related to its size or chain length, and the overall fluid environment can affect molecular diffusion. For any relatively complex compound molecule, the relaxation time is determined by intramolecular nuclear dipole interactions, which are influenced by molecular motions. The longitudinal relaxation time *T*_1_ is related to the overall molecular rotational tumbling in a solution because the frequency of tumbling must be matched with the frequency of spin transitions required for spin lattice energy transfer. Transverse relaxation explains the rapid phase dispersion of XY magnetization at a rate of 1/*T*_2_ induced by an intramolecular dynamic process in the XY plane. During this process, the longitudinal relaxation time *T*_1_ is longer than or equal to the transverse relaxation time *T*_2_. The rotational correlation time represents the time T it takes for the particle to complete one rotation arc in the solution. It is determined by the size and shape of the particles. The rate of molecular trigger was estimated using the Stokes–Einstein equation (Eq.) [20,21]:
(1)$$\frac{1}{\tau_{c}}=\frac{3KT}{4\pi \eta r_{g}^{3}}$$

According to this formula, 1/*τ_c_* is the trigger rate, *r_g_* is the radius of gyration, *η* is the viscosity of the solvent, *K* is the Boltzmann constant and *T* is the temperature in Kelvin. The theoretical model, which is a model considering both the Brownian motion of the molecule and the NMR relaxation time, was built on the basis of Bloembergen’s theory. The relaxation rate of the protons within a sample as a function of the rotational correlation time of the molecule *τ_c_* is shown below:
(2)$$\frac{1}{T_{1}}=\frac{3}{10}{\left(\frac{\mu_{0}}{2\pi }\right)}^{2}\frac{\gamma^{4}\hbar^{2}}{l^{6}}\left[\frac{\tau_{c}}{1+\omega_{0}^{2}\tau_{c}^{2}}+\frac{4\tau_{c}}{1+4\omega_{0}^{2}\tau_{c}^{2}}\right]$$
(3)$$\frac{1}{T_{2}}=\frac{3}{20}{\left(\frac{\mu_{0}}{2\pi }\right)}^{2}\frac{\gamma^{4}\hbar^{2}}{l^{6}}\left[3\tau_{c}+\frac{5\tau_{c}}{1+\omega_{0}^{2}\tau_{c}^{2}}+\frac{2\tau_{c}}{1+4\omega_{0}^{2}\tau_{c}^{2}}\right]$$

In these formulae, *ℏ* is the rationalized Planck constant, which is the Planck constant divided by 2*π*. *γ* is the permeability as the ratio of magnetic induction strength B to magnetic field strength h in a magnetic medium. *M*_0_ is the permeability over free space, where f is the 1H Larmor precession frequency, indicating *ω*_0_ = 2*π*f. l is the distance between two adjacent hydrogen nuclei located on the same compound molecule.

According to Equations (3) and (4), the downfield NMR relaxation times, *T*_1_ and *T*_2_, differ if the oil contains macromolecular structures. For rapidly moving molecules, both *T*_1_ and *T*_2_ relaxation times decrease with increasing t. For slow-moving molecules, the *T*_2_ relaxation times decrease with increasing *τ_c_*, *T*_1_ relaxation times increase with increasing *τ_c_* and *T*_1_ relaxation times are longer than *T*_2_ relaxation times. The size of the molecular aggregates and macromolecules is larger than that of isolated molecules. The reason why *T*_1_ relaxation times are longer than *T*_2_ relaxation times may be that the trigger rotation rates of these structures are low. Both relaxation time *T*_1_ or relaxation time *T*_2_ are determined by the rotational correlation time. Thus, NMR relaxation reflects molecular motion: small molecules diffuse more rapidly than large molecules. Thereby, spins on the same molecule in similar solvents have comparable relaxation times.
(4)$$\frac{T_{1}}{T_{2}}=\frac{1}{2}\frac{5{\left(1+\omega_{0}^{2}\tau_{c}^{2}\right)}^{-1}+2{\left(1+4\omega_{0}^{2}\tau_{c}^{2}\right)}^{-1}+3}{{\left(1+\omega_{0}^{2}\tau_{c}^{2}\right)}^{-1}+4{\left(1+4\omega_{0}^{2}\tau_{c}^{2}\right)}^{-1}}$$

The viscosity of a fluid sample is jointly determined by all the components in that fluid. It can be analyzed according to Equations (2) and (3), considering that the longitudinal relaxation time is equal to the transverse relaxation time at low magnetic field strength for a fluid of lower viscosity. If the *T*_1_ relaxation time is not equal to the *T*_2_ relaxation time, this indicates the presence of slower motions compared to molecular-size-related motions, which generally occurs in the presence of larger aggregated polymeric molecules in the fluid. Figure 1 demonstrates that molecular motion is accelerated (for the Larmor frequency of atoms) when *ω*_0_
*τ_c_* < 1, while for a given value, both *T*_1_ relaxation times and *T*_2_ relaxation times decrease as *τ_c_* increases. However, when the rate of molecular motion is gradually slowed, the *T*_2_ relaxation time will continue to decrease, while the *T*_1_ relaxation time will start to increase. This is most likely caused by supramolecular structures or intramolecular aggregation.

## 3. Data Analysis

Because oils have differing viscosity, the longer the carbon chain length of the oil, the higher the saturation degree, and the greater the viscosity. On the contrary, the shorter the carbon chain length of the oil, the lower the saturation degree of the oil, and the lower the viscosity of the oil. Figure 2 and Figure 3 show the measurement results for the *T*_1_-*T*_2_ distribution of eight different oils. The first type of oil was characterized by high saturation, high viscosity and a shorter relaxation time lasting under 10 ms, such as glycerin, diglycerin, etc.; the second type was resin. The *T*_1_ relaxation time of resin-containing grease was longer than the *T*_2_ relaxation time. Close interaction with other oil molecules slowed down the rotation of maltene molecules. This reaction resulted in a shorter relaxation time, such as that seen for terpineol. These results indicate that the distribution of *T*_1_-*T*_2_ is related to the physical properties and the chemical composition of oils. Therefore, the *T*_1_-*T*_2_ distribution shape can be used to identify different grease properties, as shown in Figure 2 and Figure 3.

As the oil temperature reached that what is needed for solidification, the crystals emerged, and the sample became solid (Figure 3). The gel formed a rigid network that stopped the oil from flowing. The network consisted of a fraction of the weight of the sample; at the molecular level, most oil molecules remain in the fluid state. *T*_2_ was unchanged and *T*_1_ was increased, indicating that the rotation of most oil molecules was not affected by the formation of rigid networks. Note that in this case, oil viscosity cannot be predicted from relaxation measurements alone.

The theoretical relationship between the distribution and *T*_1_, *T*_2_ and *T*_1_/*T*_2_ can be obtained from Equation (4). The results were analyzed to reveal the movement differences of each type and size of molecules in the fluid. Molecular dynamics reflects fluid. The distribution of rotation correlation time also reflects the rotation correlation time of all molecules, which can be used to analyze the molecular dynamics of complex fluids.

Therefore, oils with similar viscosity and molecular structure were selected for data analysis and processing (Table 1).

In the experiment, the transverse relaxation time rate *R*_2_ depends on the composition of different oil molecules in the sample. The Bloembergen–Purcell–Pound (BPP) method is usually used to describe the spin–spin NMR relaxation in liquids. According to the BPP theory, the spin–spin relaxation time *T*_2_ or the relaxation rate *R*_2_ (*R*_2_ = 1/*T*_2_) depends on the molecular rotation rate expressed by the rotation correlation time *τ_c_* (Figure 1), which is a characteristic parameter of the molecular rotation rate. The liquid sample *ω*_0_*τ_c_* ≪ 1 (the resonance frequency of the nuclear magnetic resonance device) and *R*_2_ increase linearly with an increase in the relevant time, as shown below:(5)$$R_{2}=10M_{2}\tau_{C}$$

In this equation, *M*_2_ is the value of the second-order matrix, which is determined by the intensity of dipole–dipole interaction between adjacent atomic nuclei. The rotation correlation time in the BPP equation can be described by the Stokes–Einstein–Debye equation:
(6)$$\tau_{c}=\frac{C_{r}\eta V}{KT}$$
(7)$$\tau_{c}\sim \eta V$$
where *V* is the effective volume of the molecule, *η* is the viscosity, *T* is the Kelvin temperature, *K* is the Boltzmann constant and is determined by experiments and *C_r_* is the fitting parameter. The Stokes–Einstein–Debye equation is usually used for homogeneous fluid in its modified form, and its molecules are described as spheres with hydrodynamics or a Stokes radius, rather than molecules.

Rotation-related motion contributes the most to the NMR relaxation spectrum, while the frequency range of the translation motion contributes less. The measurement of NMR relaxation does not consider that vibration may be due to high-amplitude frequency. The rotation correlation time of different molecular motions is a time parameter in a specific correlation function. Complex modeling is required to describe them separately. On the other hand, the rotational correlation time of molecules can be calculated by the known corresponding diffusion coefficient, which is described by the Stokes–Einstein equation:
(8)$$D_{r}=\frac{KT}{8\pi \eta R^{3}}$$
(9)$$D_{r}\sim \frac{1}{\eta R^{3}}$$
where *D_r_* is the rotational diffusion coefficient, *K* is the Boltzmann constant, *T* is the temperature, *η* is the viscosity and *R* is the Stokes molecular radius. When describing the rotational motion of molecules, it is often thought that molecules have a spherically shaped hydrodynamic radius. Considering that the effective volume of the molecule is equal to:
(10)$$V=\frac{4\pi R^{3}}{3}$$

The combination of Equations (5)–(7) gives the following results:(11)$$\tau_{c}\;\sim \;\frac{1}{D_{r}}$$

Therefore, it is clear that the rotation correlation time depends on the corresponding diffusion coefficient (Equation (10)). Equation (10) is a widely used ratio for calculating the NMR relaxation correlation time, because the rotational molecular motion is the main contributor to NMR relaxation. In general, molecules with different shapes can be characterized by hydrodynamic radii. In order to correlate the rotational diffusion coefficient with the molecular weight, known ratios can be used to describe the self-diffusion and self-behavior of the untwisted polymer chain. For molecules, there is a direct dependency between these two parameters:(12)$$D_{self}\sim M_{w}^{-\alpha }\;<\;M_{w}<M_{wc}$$
where *M_w_*_c_ is the entanglement coupling molecular weight, and *α* is the coefficient, equal to 1. By combining Equations (6), (10), and (12), we can clearly see the linear relationship between *R*_2_ and molecular weight.
(13)$$\frac{R_{2}}{\eta }\sim M_{w}$$

It is easy to assume that in the case of unbranched hydrocarbons with similar chemical organization, the molecular weight is proportional to the molecular size, which can be evaluated by CL. Considering the dependence of *R*_2_ on the molecular weight *M_w_* (Equation (13)), we can assume that *R*_2_ is linearly dependent on CL:
(14)$$\frac{R_{2}}{\eta }\sim \;\mathrm{CL}\;$$

The decay of the CPMG pulse-echo sequence of all oil samples was measured, and then the spin–spin relaxation rate *R*_2_ was determined using single-exponential function fitting. The obtained *R*_2_ values are provided in Table 1. The carbon chain length CL is plotted as a function of the relaxation rate in Figure 4, showing a linear relationship. By applying a linear fit to the dataset, the model can be calculated as follows:
(15)$${\mathrm{CL}}_{NMR}=a\;R_{2}+b$$
where *a* = −52.998 and *b* = −1.5597.

In this study, the correlation between carbon chain length and relaxation rates was obtained, showing good consistency with previous results. We can use this to obtain the carbon chain length by using LF-NMR. Grease was selected due to its properties, including a temperature lower than room temperature. However, in addition to the main components, oils can have higher carbon chain lengths (more than 20). Therefore, the carbon chain length of general vegetable oils is generally longer than that used in this study. From Equation (13) and our experimental results, it can be seen that the relaxation rate of oil molecules has a linear relationship with the length of the carbon chain. It is clear that this relationship applies to longer molecules.

## 4. Experiment

The oil samples used in this experiment were various vegetable oils mainly composed of glycerides and fatty acids. Oil samples kept at a constant temperature were loaded into the test tubes and labeled. These were capped and sealed to avoid volatilization of the light hydrocarbon components during NMR relaxometry, which ensured accurate, effective and error-free relaxation time measurement. Oil and fat samples with different carbon chain lengths were prepared by Shanghai Macklin Biochemical Technology Co. (Shanghai, China) The oils and fats in this study were selected based on their properties, and they were maintained at a temperature below their melting point, 25 °C, for measurement at room temperature. Their properties (molecular weight, state and molecular carbon chain length) are given in Table 2. The viscosity of the samples was measured at 20 °C.

Table 1 was calculated using the hard-pulse CPMG (Carr–Parcell–Meiboom–Gill) sequence, and the echo time and number of echoes were set. The instrument used for relaxometry measurements in this experiment was MicroMR from the Niumag company, with a main frequency of 23 MHz. In this experiment, 16 repeated scans were carried out because the signal intensity of oil was relatively strong compared to the moisture signal. The sampling frequency setting was set to approximately 250 kHz to obtain more effective signals in grease. The parameters used are as follows: repetitive sampling latency TR, 10 s; echo time TE, 0.3 ms; the number of echoes was around 1000 and main value SF of the RF signal frequency, 23 MHz. Measurements for all experiments were performed at room temperature (25 °C).

## 5. Conclusions

We used NMR relaxation to measure vegetable oil, collected and analyzed NMR signals of more than 20 different kinds of vegetable oil and essential oil samples, obtained the oil samples’ transverse relaxation times and the rate of oil samples and determined the possible interaction of these molecules by inferring and analyzing the relationship between the length of the oil carbon chain and the transverse relaxation time rate. NMR relaxation reflects molecular motion. Small molecules diffuse farther and rotate faster than large molecules, indicating that all spins on the same molecule in similar solvents have a considerable relaxation time. The results of this study show that the rotation-related time distribution is directly related to the molecular size distribution and molecular dynamics in oil solutions. The rotation correlation time of oil also reflects the interaction between molecules and the viscosity of the oil solution. We obtained the correlation between the NMR results and the molecular carbon chain length, which can be used to determine the carbon chain length.

## Figures and Tables

**Figure 1 molecules-28-00197-f001:**
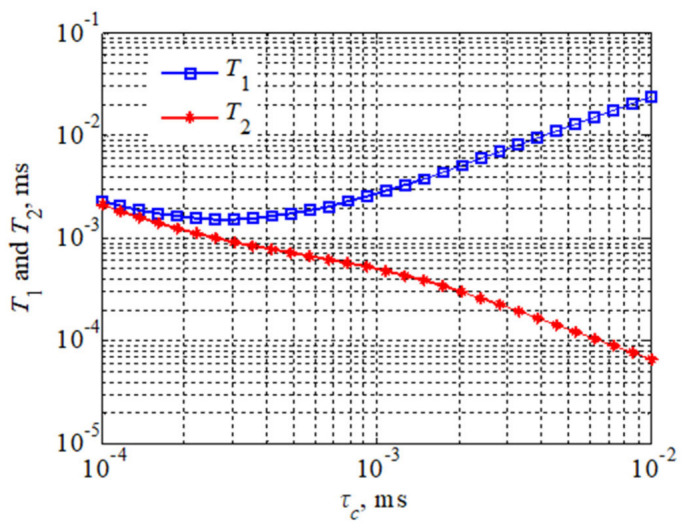
Theoretical plots of *T*_1_ and *T*_2_ relaxation times versus Larmor frequency and correlation times.

**Figure 2 molecules-28-00197-f002:**
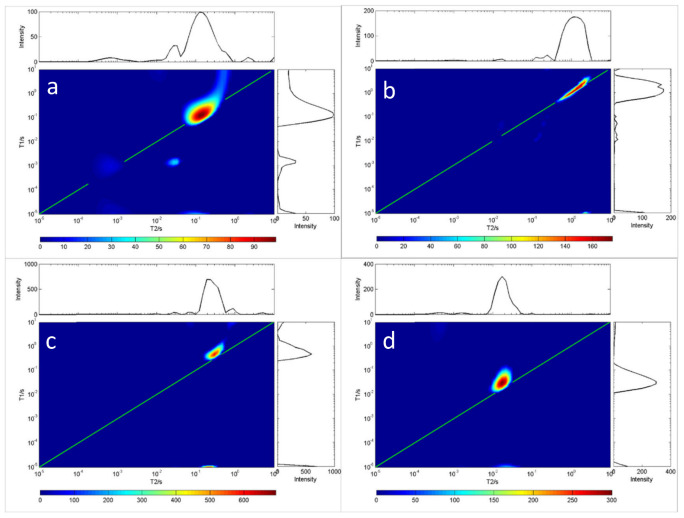
*T*_1_-*T*_2_ distribution of four oil samples at room temperature (20 °C). In each chart, the green line indicates the *T*_1_ = *T*_2_ line. (**a**) C_18_H_34_O_2_, (**b**) C_24_H_38_O_4_, (**c**) C_12_H_20_O_2_, (**d**) C_3_H_8_O_3_.

**Figure 3 molecules-28-00197-f003:**
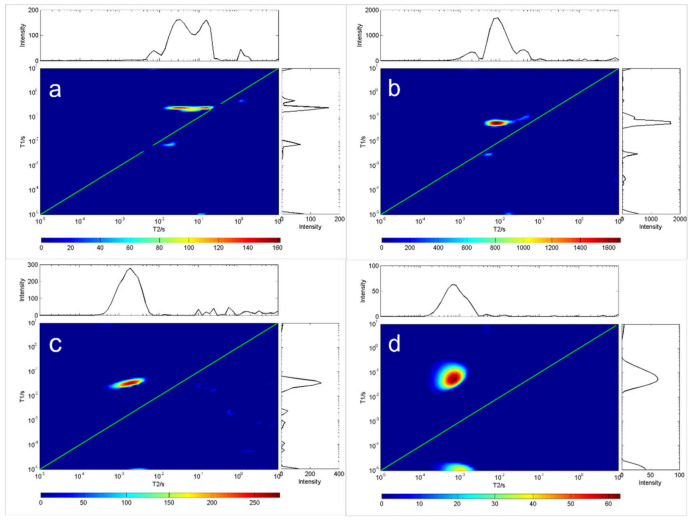
*T*_1_-*T*_2_ distribution of four oil samples at room temperature (20 °C). In each chart, the green line indicates the *T*_1_ = *T*_2_ line. (**a**) C_21_H_42_O_4_, (**b**) C_24_H_47_NO_4_, (**c**) C_6_H_14_O_5_, (**d**) C_30_H_62_O_21_.

**Figure 4 molecules-28-00197-f004:**
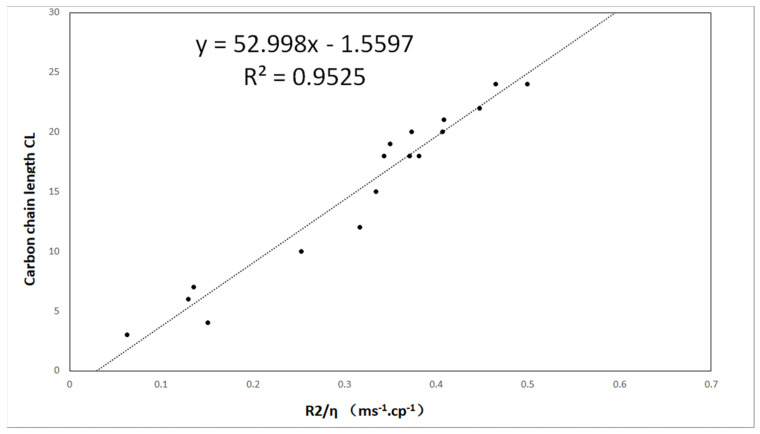
The carbon chain length CL is plotted as a function of *R*_2_/*η*. The dotted line is fitted by the linear model.

**Table 1 molecules-28-00197-t001:** Experimental values of carbon chain length and spin–spin relaxation rate of oils.

Formula	Name	Chain Length	Status	*T*_2_ (ms)	*R*_2_ (ms^−1^)	*R*_2_/*η* (ms^−1^cp^−1^)	*η* (cp, 20 °C)
C_3_H_8_O_3_	Glycerol	3	liquid	10	93.95189	0.062635	1500
C_4_H_8_O_3_	Glyceraldehyde	4	liquid	451	2.214346	0.150636	14.7
C_6_H_12_O_3_	Diglycerol	6	liquid	1.3	725.8648	0.129619	5600
C_7_H_12_ClN_3_O_2_	Ketal glycerol	7	liquid	520	1.921587	0.135323	14.2
C_10_H_18_O	Terpineol	10	liquid	50	19.69276	0.252471	78
C_12_H_20_O_6_	Tripropionin	12	liquid	362	2.755334	0.316705	8.7
C_15_H_26_O_6_	Tributyrin	15	liquid	325	3.073538	0.33408	9.2
C_18_H_34_O_2_	Oleic acid	18	liquid	153	6.524779	0.370726	17.6
C_18_H_37_N	Oleylamine	18	liquid	404	2.470074	0.343066	7.2
C_18_H_36_O	Oleyl alcohol	18	liquid	171	5.838505	0.381103	15.32
C_19_H_36_O_2_	methyl oleate	19	liquid	520	1.921587	0.34938	5.5
C_20_H_36_O_2_	ethyl linoleate	20	liquid	503	1.985095	0.406782	4.88
C_20_H_38_O_2_	Ethyl Oleate	20	liquid	520	1.920025	0.37282	5.15
C_21_H_20_O_6_	Curcuma oil	21	liquid	480	2.082722	0.408377	5.1
C_22_H_42_O_2_	Butyl oleate	22	liquid	465	2.147456	0.447387	4.8
C_24_H_38_O_4_	Perilla leaf oil	24	liquid	1266	0.789854	0.46462	1.7
C_24_H_47_NO_4_	Triethanolamine oleic acid soap	24	liquid	8.5	117.3367	0.499305	235

**Table 2 molecules-28-00197-t002:** Oils and fats used for sample preparation.

Formula	Name	Ar	Chain Length	Status	Purity	H (cp, 20 °C)
C_3_H_8_O_3_	Glycerol	92	3	Liquid, viscous	98%	1500
C_4_H_8_O_3_	Glyceraldehyde	104	4	Liquid	98%	14.7
C_6_H_14_O_5_	Diglycerol	166	6	Liquid, viscous	80%	5600
C_6_H_12_O_3_	Ketal glycerol	132	6	Liquid	≥95%	N/A
C_7_H_12_ClN_3_O_2_	Clove oil ^1^	205.5	7	Liquid	AR, ≥80%	14.2
C_10_H_16_	Turpentine	136	10	Liquid	AR	2.1
C_10_H_18_O	Terpineol ^2^	154	10	Liquid	AR	68
C_12_H_20_O_6_	Glyceryl tripropanoate	260	12	Liquid	≥95%	8.7
C_12_H_20_O_2_	Turpentine acetate	196	12	Liquid	≥97%	N/A
C_15_H_26_O_6_	Tributyrin	302	15	Liquid	98%	9.2
C_18_H_34_O_2_	Oleic acid	282	18	Liquid	AR	17.6
C_18_H_37_N	Oleylamine	267	18	Solid	80–90%	7.2
C_18_H_36_O	Oleyl alcohol	268	18	Liquid	80–85%	15.32
C_19_H_36_O_2_	Methyl oleate	296	19	Liquid	99%	5.5
C_20_H_36_O_2_	Linoleic acid ethyl ester	308	20	Liquid	≥97%	4.88
C_20_H_38_O_2_	Ethyl oleate	310	20	Liquid	75%	5.15
C_21_H_20_O_6_	Ginger butter	368	21	Liquid	≥98%	N/A
C_21_H_42_O_4_	Propylene glycol oleate	358	21	Liquid	95%	5.1
C_22_H_42_O_2_	Butyl oleate	338	22	Liquid	AR	4.8
C_24_H_38_O_4_	Perilla leaf oil	390	24	Liquid	≥55%	1.7
C_24_H_47_NO_4_	Triethanolamine oleic acid soap	413	24	Liquid	92%	235
C_27_H_50_O_6_	Trioctanoin	470	27	Liquid	≥99%	N/A
C_30_H_62_O_21_	Decaprenylglycerol	758	30	Solid	98%	N/A
C_38_H_46_N_2_O_8_	Cocamine	658	38	Solid	98%	N/A
C_39_H_76_O_5_	Distearate	624	39	Solid	97%	N/A
C_39_H_74_O_6_	Glycerol trilaurate	638	39	Solid	98%	N/A
C_45_H_76_O_2_	Linoleate cholesteryl ester	648	45	Solid	95%	N/A
C_51_H_98_O_6_	Tripalmitin	806	51	Solid	98%	N/A
C_57_H_110_O_6_	Tristearin	890	57	Solid	98%	N/A

^1^ Essential oil ^2^ Essential oil, mixtures of isomers.

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
