# Peer review of "Molecular Dynamics and Chain Length of Edible Oil Using Low-Field Nuclear Magnetic Resonance"

_molecules, 2022, doi:10.3390/molecules28010197_

Round 1

Reviewer 1 Report (Previous Reviewer 1)

I managed to check the manuscript. The authors impoved significantly the quality of the manuscript and adressed all my comments and recomandations. There are still small things I do not understand as for example, the authors did not provided an adequate answer why table 1 is important and about why T1-T2 correlations only for selected samples. But given that I do not believe that those answers will not further strongly improve the quality of the manuscript. I suggest thus to accept it in the current form.

Author Response

Dear Reviewer,

Thank you for your comments concerning our manuscript entitled “Molecular dynamics and Chain length of edible oil by low-field nuclear magnetic resonance” (ID: molecules-2112583). We quite appreciate your favorite consideration and the your insightful comments. We did a min revision to the manuscript according to the your comments, and found these comments are very helpful.

If the measured sample of oil mixture contains macromolecules, the phenomenon of macromolecular aggregation will occur. This phenomenon can be easily measured by T1-T2 measurement, and the result is similar to Fig. 3cd. But more research is needed to for quantitative analysis.

We hope this revision can make our manuscript more acceptable. Revised portion are marked up using the “Track Changes” function in the manuscript. Please feel free to contact us if there is any questions.

Sincerely yours

Reviewer 2 Report (New Reviewer)

The manuscript by Zijian Jia and Can Liang demonstrate the uses of low-field NMR for predicting the size of edible oil molecules (relative molecular mass, carbon chain length).

The manuscript is well written, however few scientific news are identified in the paper. The theoretical content of the methods are fine, but the results obtained are not really news, as many papers already have demonstrated that for general cases.

If the intention of the journal is publishing a brief description of LowField-NMR methods to link relaxometry and molecular dynamics and chain length in edible oils, the scientific content is ok, but there are few scientific news.

Observations:

Line 80  -  “fatty acid and organic..”

Line 112  -  μ0  

Line 113  -  w0  

Line 118  -  tc      (two times..)

Line 129     EQS was not defined

Line 134     “ Larmor ”

Line 135  -  w0   “ and  tc     

Line 141     “ Larmor ”

Line 207  -  w0  

Line 208  -  R2  

Line 220  -  In line 103, “ k “ was the Boltzmann constant, and in line 220 “ KB  “. Define one.

Line 290 –  “NMR relaxation”

Line 294 –  “oil solutions ”

Author Response

Dear Reviewer,

Thank you for your comments concerning our manuscript entitled “Molecular dynamics and Chain length of edible oil by low-field nuclear magnetic resonance” (ID: molecules-2112583). We quite appreciate your favorite consideration and the your insightful comments. We did a min revision to the manuscript according to the your comments, and found these comments are very helpful.

We have revised all the symbol error. We hope this revision can make our manuscript more acceptable. Revised portion are marked up using the “Track Changes” function in the manuscript. Please feel free to contact us if there is any questions.

Sincerely yours,

This manuscript is a resubmission of an earlier submission. The following is a list of the peer review reports and author responses from that submission.

Round 1

Reviewer 1 Report

Molecular Dynamics and Chain Length of Edible Oil by Low-Field Nuclear Magnetic Resonance

by Zijian Jia and Can Liang

The authors use low-field NMR relaxometry to analyse the molecular dynamics of edible oils. While the topic is interesting and the abstract is promising, the manuscript has many week points and thus I am not able to support the publication of the manuscript.

1. A major issue is that the English of the manuscript is of very low level. Starting already with the abstract (line 16), there are alone standing words, which should be sentences. Along the same issue: introduction line 40-41, experimental lines 146-147etc.

2. Introduction: The authors state that “Compared with high-field NMR, LF-NMR is low in both capital and time cost.” It is not clear what it is the meaning of lower time cost.

3. Introduction: The authors state that “The determination of carbon chain length is very important for liquid fuels because it is one of the key parameters determining their quality and performance.” The authors should add references to support this statement.

4. One except in the section “Materials and methods” to read more details about the materials but this whole section is focusing on the NMR theory. The name of this section should be changed to reflect its content.

5. In the experimental section is stated that “Samples with different average carbon chain lengths were prepared as different mixtures of several common vegetable oil oils and fats”. It is not clear where these samples are used in the current study.

6. In the experimental section is stated that “NMR spectra of oils are characterized by overlapping signals of a wide variety of fatty acids, which come from among different triglyceride combinations”. This is a correct statement but is not clear why it is necessary.

7. The authors should include details about the equipment used to conduct the measurements. Is this an equipment used for conducting spectroscopy/relaxometry measurements or it is able to conduct only relaxometry?

8. Experimental: use the whole name for CPMG and add reference.

9. Experimental: it is not clear why 1000 echoes if only 600 180 pulses were used

10.  Experimental: the authors state that an echo time of 6 ms was used. In the T1-T2 data, the T2 values of most shown samples are in the range of 1 ms or lower. Thus, it is the question about the reliability of the conducted measurements.

11.  Table 1 shows a huge number of oils for sample preparations. It is not clear how the sample preparation is done.

12.  In section 4, it is stated that the viscosity is of the different oils is different. Viscosity plays an important role in the studies that the authors present and thus they should include the values of the viscosities for the studied oils.

13.  Section data analysis: last paragraph from line 172 is not belonging to the manuscript.

14.  It is also not clear why time consuming T1-T2 measurements were conducted. Why not simply T1 and T2 measurements?

15.   In the abstract it is stated that relaxation time distributions will be used. No information about the distribution is reported.

16.  In the T1-T2 data, the T2 values are lower than 1 s for all reported samples, including the C6H12O3 for which in table 2 is reported of being around 3.6 s. which data are now true?

17.  More, the T2 data from Figure 2 and Figure 3 indicate that T2 gets shorter with increasing the number of carbons and it gets in the region of ms. How comes than that the T2 values reported in Table 2 are all in seconds? Which is the error of these measurements?

18.  Figure 4: unit of the relaxation rate is missing. More, the dispersion of the data is high and thus is difficult to say if the correlation is linear.

19.  Using the data from Figure 4, the authors state that “the carbon chain length of oil mixture has been determined”. This is not a determination but a try to get a correlation, which eventually can be used for such a determination purpose.

Author Response

Dear Reviewer,

Thank you very much for your review comments. Your comments inspired me to revise the analysis method of this article. Because I used fuel alkanes for experiments and analysis before, the viscosity is linearly related to the carbon chain length. Even mixed alkanes viscosities are linearly related to average chain length. Therefore, the influence of viscosity on the results was not considered at the beginning, resulting in unsatisfactory results. After you reminded me to introduce viscosity in the analysis, the results are very good. But oils are still complex, and molecular dynamics and viscosity analysis of mixed oils is needed in the next research.

  1. A major issue is that the English of the manuscript is of very low level. Starting already with the abstract (line 16), there are alone standing words, which should be sentences. Along the same issue: introduction line 40-41, experimental lines 146-147etc.

This have been revised by native speaker.

  1. Introduction: The authors state that “Compared with high-field NMR, LF-NMR is low in both capital and time cost.” It is not clear what it is the meaning of lower time cost.

Low-field NMR instruments can be made portable. Testing can be done on site rather than in a fixed laboratory. I have add the information and reference. See line 50-51.

  1. Introduction: The authors state that “The determination of carbon chain length is very important for liquid fuels because it is one of the key parameters determining their quality and performance.” The authors should add references to support this statement.

I have add the information and reference. See line 69-70.

  1. One except in the section “Materials and methods” to read more details about the materials but this whole section is focusing on the NMR theory. The name of this section should be changed to reflect its content.

This have been revised.

  1. In the experimental section is stated that “Samples with different average carbon chain lengths were prepared as different mixtures of several common vegetable oil oils and fats”. It is not clear where these samples are used in the current study.

I am so sorry to have this mistake. I have made the samples for this research. For the NMR data is so complicated, I have used it in next research for oil molecules size. I have revised it.

  1. In the experimental section is stated that “NMR spectra of oils are characterized by overlapping signals of a wide variety of fatty acids, which come from among different triglyceride combinations”. This is a correct statement but is not clear why it is necessary.

I am sorry to make you misunderstand. I want to express that for mixed oil samples the relaxation times are more appropriate rather than spectra. This sentence is really inappropriate here. I have revised it.

  1. The authors should include details about the equipment used to conduct the measurements. Is this an equipment used for conducting spectroscopy/relaxometry measurements or it is able to conduct only relaxometry?

It is only for relaxometry. I have add the experiment information in line 154-156.

  1. Experimental: use the whole name for CPMG and add reference.

I have add the experiment information in line 153-154.

  1. Experimental: it is not clear why 1000 echoes if only 600 180 pulses were used

I have check all the experiment information and correct them. See in line156-161.

  1.  Experimental: the authors state that an echo time of 6 ms was used. In the T1-T2 data, the T2 values of most shown samples are in the range of 1 ms or lower. Thus, it is the question about the reliability of the conducted measurements.

I have checked all experimental data and parameters and have corrected previous errors. There was a decimal point error in the previous data export.

  1.  Table 1 shows a huge number of oils for sample preparations. It is not clear how the sample preparation is done.

These samples were purchased from commercial companies. They are made by different ways, such as distillation, chemical leaching and compounding.

  1.  In section 4, it is stated that the viscosity is of the different oils is different. Viscosity plays an important role in the studies that the authors present and thus they should include the values of the viscosities for the studied oils.

I have add all the information in table 1.

  1.  Section data analysis: last paragraph from line 172 is not belonging to the manuscript.

I have revised them.

  1.  It is also not clear why time consuming T1-T2 measurements were conducted. Why not simply T1 and T2 measurements?

That is because T1 and T2 measurements only give 1D information, For simple samples, a simple 1D measurement is sufficient. However, for complex samples, different T1 signals may overlap in the T2 direction, making it impossible to analyze the composition of the sample.

  1.   In the abstract it is stated that relaxation time distributions will be used. No information about the distribution is reported.

For pure fluids to validate this method, no relaxation distribution is required. The relaxation distribution is more desirable for mixed fluids. I have revised them.

  1.  In the T1-T2 data, the T2 values are lower than 1 s for all reported samples, including the C6H12O3 for which in table 2 is reported of being around 3.6 s. which data are now true?

I have checked all experimental data and parameters and have corrected previous errors. There was a decimal point error in the previous data export.

  1.  More, the T2 data from Figure 2 and Figure 3 indicate that T2 gets shorter with increasing the number of carbons and it gets in the region of ms. How comes than that the T2 values reported in Table 2 are all in seconds? Which is the error of these measurements?

I have checked all experimental data and parameters and have corrected previous errors. There was a decimal point error in the previous data export.

  1.  Figure 4: unit of the relaxation rate is missing. More, the dispersion of the data is high and thus is difficult to say if the correlation is linear.

I have revised the method used to include the effect of viscosity. Because the samples I used before are alkanes, the viscosity is directly related to the average molecular weight and carbon chain length, so it can be ignored in the formula. With your reminder, I found that this rule does not fully apply to samples such as oleic acid. After introducing the viscosity parameter, this method has been corrected.

  1.  Using the data from Figure 4, the authors state that “the carbon chain length of oil mixture has been determined”. This is not a determination but a try to get a correlation, which eventually can be used for such a determination purpose.

I have modified my expression.

Zijian Jia

Reviewer 2 Report

Jia and Liang studied the molecular dynamics and chain length of edibe oil by LFNMR. It is a novelty study and provided a new analysis method for molecular dynamics using LFNMR. Results showed that LFNMR obtain information on the composition, macromolecular aggregation, and molecular dynamics of complex fluids. This study will be helpful to explore the application of LFNMR. I think this manuscript could be acceptable for publication after minor revision.

1.     In the part of introduction, the edible oil information should be supplied and authors should emphasis the reason of study edible oil by LFNMR.

2.     Figure 4 should be redone because the quality was too low.

3.     The conclusion should be rewritten and concise.

Author Response

Dear Reviewer,

Thank you very much for your review comments. Your comments inspired me to revise the analysis method of this article.

  1. In the part of introduction, the edible oil information should be supplied and authors should emphasis the reason of study edible oil by LFNMR.

I have add the information in introduction.

  1. Figure 4 should be redone because the quality was too low.

I have upload the high quality Figure.

  1. The conclusion should be rewritten and concise.

I have rewrite it.

Zijian Jia

USST